# Adaptive Averaging in Accelerated Descent Dynamics

**Walid Krichene** *
UC Berkeley
walid@eecs.berkeley.edu

**Alexandre M. Bayen**
UC Berkeley
bayen@berkeley.edu

**Peter L. Bartlett**
UC Berkeley and QUT
bartlett@cs.berkeley.edu

## Abstract

We study accelerated descent dynamics for constrained convex optimization. This dynamics can be described naturally as a coupling of a dual variable accumulating gradients at a given rate $\eta(t)$, and a primal variable obtained as the weighted average of the mirrored dual trajectory, with weights $w(t)$. Using a Lyapunov argument, we give sufficient conditions on $\eta$ and $w$ to achieve a desired convergence rate. As an example, we show that the replicator dynamics (an example of mirror descent on the simplex) can be accelerated using a simple averaging scheme.

We then propose an adaptive averaging heuristic which adaptively computes the weights to speed up the decrease of the Lyapunov function. We provide guarantees on adaptive averaging in continuous-time, prove that it preserves the quadratic convergence rate of accelerated first-order methods in discrete-time, and give numerical experiments to compare it with existing heuristics, such as adaptive restarting. The experiments indicate that adaptive averaging performs at least as well as adaptive restarting, with significant improvements in some cases.

## 1   Introduction

We study the problem of minimizing a convex function $f$ over a feasible set $\mathcal{X}$, a closed convex subset of $E = \mathbb{R}^n$. We will assume that $f$ is differentiable, that its gradient $\nabla f$ is a Lipschitz function with Lipschitz constant $L$, and that the set of minimizers $S = \arg\min_{x \in \mathcal{X}} f(x)$ is non-empty. We will focus on the study of continuous-time, first-order dynamics for optimization. First-order methods have seen a resurgence of interest due to the significant increase in both size and dimensionality of the data sets typically encountered in machine learning and other applications, which makes higher-order methods computationally intractable in most cases. Continuous-time dynamics for optimization have been studied for a long time, e.g. [6, 9, 5], and more recently [20, 2, 1, 3, 11, 23], in which a connection is made between Nesterov's accelerated methods [14, 15] and a family of continuous-time ODEs. Many optimization algorithms can be interpreted as a discretization of a continuous-time process, and studying the continuous-time dynamics is useful for many reasons: The analysis is often simpler in continuous-time, it can help guide the design and analysis of new algorithms, and it provides intuition and insight into the discrete process. For example, Su et al. show in [20] that Nesterov's original method [14] is a discretization of a second-order ODE, and use this interpretation to propose a restarting heuristic which empirically speeds up the convergence. In [11], we generalize this approach to the proximal version of Nesterov's method [15] which applies to constrained convex problems, and show that the continuous-time ODE can be interpreted as coupled dynamics of a dual variable $Z(t)$ which evolves in the dual space $E^*$, and a primal variable $X(t)$ which is obtained as the weighted average of a non-linear transformation of the dual trajectory. More precisely,

$$
\begin{cases}
\dot{Z}(t) = -\frac{t}{r}\nabla f(X(t)) \\
X(t) = \frac{\int_0^t \tau^{r-1}\nabla\psi^*(Z(\tau))d\tau}{\int_0^t \tau^{r-1}d\tau} \\
X(0) = \nabla\psi^*(Z(0)) = x_0,
\end{cases}
$$

where $r \geq 2$ is a fixed parameter, the initial condition $x_0$ is a point in the feasible set $\mathcal{X}$, and $\nabla\psi^*$ is a Lipschitz function that maps from the dual space $E^*$ to the feasible set $\mathcal{X}$, which we refer to as the mirror map (such a function can be constructed using standard results from convex analysis, by taking the convex conjugate of a strongly convex function $\psi$ with domain $\mathcal{X}$; see the supplementary material for a brief review of the definition and basic properties of mirror maps). Using a Lyapunov argument, we show that the solution trajectories of this ODE exhibit a quadratic convergence rate, i.e. if $f^\star$ is the minimum of $f$ over the feasible set, then $f(X(t)) - f^\star \leq C/t^2$ for a constant $C$ which depends on the initial conditions. This formalized an interesting connection between acceleration and averaging, which had been observed in [8] in the special case of unconstrained quadratic minimization.

A natural question that arises is whether different averaging schemes can be used to achieve the same rate, or perhaps faster rates. In this article, we provide a positive answer. We study a broad family of Accelerated Mirror Descent (AMD) dynamics, given by

$$\mathrm{AMD}_{w,\eta} \begin{cases} \dot{Z}(t) = -\eta(t)\nabla f(X(t)) \\[2mm] X(t) = \dfrac{X(t_0)W(t_0) + \int_{t_0}^t w(\tau)\nabla\psi^*(Z(\tau))d\tau}{W(t)}, \text{ with } W(t) = \int_0^t w(\tau)d\tau \\[2mm] X(t_0) = \nabla\psi^*(Z(t_0)) = x_0, \end{cases} \tag{1}$$

parameterized by two positive, continuous weight functions $w$ and $\eta$, where $w$ is used in the averaging and $\eta$ determines the rate at which $Z$ accumulates gradients. This is illustrated in Figure 1. In our formulation we choose to initialize the ODE at $t_0 > 0$ instead of 0 (to guarantee existence and uniqueness of a solution, as discussed in Section 2). We give a unified study of this ODE using an appropriate Lyapunov function, given by

$$L_r(X, Z, t) = r(t)(f(X) - f^\star) + D_{\psi^*}(Z, z^\star), \tag{2}$$

where $D_{\psi^*}$ is the Bregman divergence associated with $\psi^*$ (a non-negative function defined on $E^* \times E^*$), and $r(t)$ is a desired convergence rate (a non-negative function defined on $\mathbb{R}_+$). By construction, $L_r$ is a non-negative function on $\mathcal{X} \times E^* \times \mathbb{R}_+$. If $t \mapsto L_r(X(t), Z(t), t)$ is a non-increasing function for all solution trajectories $(X(t), Z(t))$, then $L_r$ is said to be a Lyapunov function for the ODE, in reference to Aleksandr Mikhailovich Lyapunov [12]. We give in Theorem 2 a sufficient condition on $\eta, w$ and $r$ for $L_r$ to be a Lyapunov function for $\mathrm{AMD}_{w,\eta}$, and show that under these conditions, $f(X(t))$ converges to $f^\star$ at the rate $1/r(t)$.

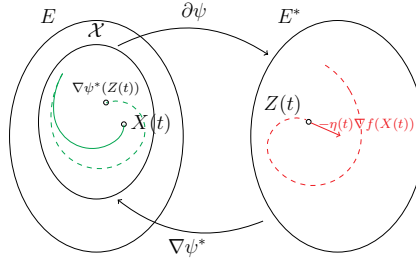

Figure 1: Illustration of $\mathrm{AMD}_{w,\eta}$. The dual variable $Z$ evolves in the dual space $E^*$, and accumulates negative gradients at a rate $\eta(t)$, and the primal variable $X(t)$ (green solid line) is obtained by averaging the mirrored trajectory $\{\nabla\psi^*(Z(\tau)), \ \tau \in [t_0, t]\}$ (green dashed line), with weights $w(\tau)$.

In Section 3, we give an equivalent formulation of $\mathrm{AMD}_{w,\eta}$ written purely in the primal space. We give several examples of these dynamics for simple constraint sets. In particular, when the feasible set is the probability simplex, we derive an accelerated version of the replicator dynamics, an ODE that plays an important role in evolutionary game theory [22] and viability theory [4].

Many heuristics have been developed to empirically speed up the convergence of accelerated methods. Most of these heuristics consist in restarting the ODE (or the algorithm in discrete time) whenever a simple condition is met. For example, a gradient restart heuristic is proposed in [17], in which the algorithm is restarted whenever the trajectory forms an acute angle with the gradient (which intuitively indicates that the trajectory is not making progress), and a speed restarting heuristic is proposed in [20], in which the ODE is restarted whenever the speed $\|\dot{X}(t)\|$ decreases (which intuitively indicates that progress is slowing). These heuristics are known to empirically improve

the speed of convergence, but provide few guarantees. For example, the gradient restart in [17] is only studied for unconstrained quadratic problems, and the speed restart in [20] is only studied for unconstrained strongly convex problems. In particular, it is not guaranteed (to our knowledge) that these heuristics preserve the original convergence rate of the non-restarted method, when the objective function is not strongly convex. In Section 4, we propose a new heuristic that provides such guarantees, and that is based on a simple idea for adaptively computing the weights $w(t)$ along the solution trajectories. The heuristic simply decreases the time derivative of the Lyapunov function $L_r(X(t), Z(t), t)$ whenever possible. Thus it preserves the $1/r(t)$ convergence rate. Other adaptive methods have been applied to convex optimization, such as Adagrad [7] and Adam [10], which adapt the learning rate in first-order methods, by maintaining moment estimates of the observed gradients. They are particularly well suited to problems with sparse gradients. While these methods are similar in spirit to adaptive averaging, they are not designed for accelerated methods. In Section 5, we give numerical experiments in which we compare the performance of adaptive averaging and restarting. The experiments indicate that adaptive averaging compares favorably in all of the examples, and gives a significant improvement in some cases. We conclude with a brief discussion in Section 6.

## 2   Accelerated mirror descent with generalized averaging

We start by giving an equivalent form of $\mathrm{AMD}_{w,\eta}$, which we use to briefly discuss existence and uniqueness of a solution. Writing the second equation as $X(t)W(t) - X(t_0)W(t_0) = \int_{t_0}^t w(\tau)\nabla\psi^*(Z(\tau))d\tau$, then taking the time-derivative, we have

$$\dot{X}(t)W(t) + X(t)w(t) = w(t)\nabla\psi^*(Z(t)).$$

Thus the ODE is equivalent to

$$\mathrm{AMD}'_{w,\eta}\begin{cases} \dot{Z}(t) = -\eta(t)\nabla f(X(t)) \\ \dot{X}(t) = \frac{w(t)}{W(t)}(\nabla\psi^*(Z(t)) - X(t)) \\ X(t_0) = \nabla\psi^*(Z(t_0)) = x_0. \end{cases}$$

The following theorem guarantees existence and uniqueness of the solution.

**Theorem 1.** *Suppose that $W(t_0) > 0$. Then $\mathrm{AMD}_{w,\eta}$ has a unique maximal (i.e. defined on a maximal interval) solution $(X(t), Z(t))$ that is $C^1([t_0, +\infty))$. Furthermore, for all $t \geq t_0$, $X(t)$ belongs to the feasible set $\mathcal{X}$.*

*Proof.* Recall that, by assumption, $\nabla f$ and $\nabla\psi^*$ are both Lipschitz, and $w, \eta$ are continuous. Furthermore, $W(t)$ is non-decreasing and continuous, as the integral of a non-negative function, thus $w(t)/W(t) \leq w(t)/W(t_0)$. This guarantees that on any finite interval $[t_0, T]$, the functions $\eta(t)$ and $w(t)/W(t)$ are bounded. Therefore, $-\eta(t)\nabla f(X)$ and $\frac{w(t)}{W(t)}(\nabla\psi^*(Z) - X)$ are Lipschitz functions of $(X, Z)$, uniformly in $t \in [t_0, T]$. By the Cauchy-Lipschitz theorem (e.g. Theorem 2.5 in [21]), there exists a unique $C^1$ solution defined on $[t_0, T]$. Since $T$ is arbitrary, this defines a unique solution on all of $[t_0, +\infty)$. Indeed, any two solutions defined on $[t_0, T_1)$ and $[t_0, T_2)$ with $T_2 > T_1$ coincide on $[t_0, T_1)$. Finally, feasibility of the solution follows from the fact that $\mathcal{X}$ is convex and $X(t)$ is the weighted average of points in $\mathcal{X}$, specifically, $x_0$ and the set $\{\nabla\psi^*(Z(\tau)), \tau \in [t_0, t]\}$. $\qquad\square$

Note that in general, it is important to initialize the ODE at $t_0$ and not $0$, since $W(0) = 0$ and $w(t)/W(t)$ can diverge at $0$, in which case one cannot apply the Cauchy-Lipschitz theorem. It is possible however to prove existence and uniqueness with $t_0 = 0$ for some choices of $w$, by taking a sequence of Lipschitz ODEs that approximate the original one, as is done in [20], but this is a technicality and does not matter for practical purposes.

We now move to our main result for this section. Suppose that $r$ is an increasing, positive differentiable function on $[t_0, +\infty)$, and consider the candidate Lyapunov function $L_r$ defined in (2), where the Bregman divergence term is given by

$$D_{\psi^*}(z, y) := \psi^*(z) - \psi^*(y) - \langle \nabla\psi^*(y), z - y \rangle,$$

and $z^\star$ is a point in the dual space such that $\nabla\psi^*(z^\star) = x^\star$ belongs to the set of minimizers $S$. Let $(X(t), Z(t))$ be the unique maximal solution trajectory of $\mathrm{AMD}_{w,\eta}$.

Taking the derivative of $t \mapsto L_r(X(t), Z(t), t) = r(t)(f(X(t)) - f^\star) + D_{\psi^*}(Z(t), z^\star)$, we have

$$\frac{d}{dt}L_r(X(t), Z(t), t) = r'(t)(f(X(t)) - f^\star) + r(t)\left\langle \nabla f(X(t)), \dot{X}(t) \right\rangle + \left\langle \dot{Z}(t), \nabla \psi^*(Z(t)) - \nabla \psi^*(z^\star) \right\rangle$$

$$= r'(t)(f(X(t)) - f^\star) + r(t)\left\langle \nabla f(X(t)), \dot{X}(t) \right\rangle + \left\langle -\eta(t)\nabla f(X(t)), X(t) + \frac{W(t)}{w(t)}\dot{X}(t) - x^\star \right\rangle$$

$$\leq (f(X(t)) - f^\star)(r'(t) - \eta(t)) + \left\langle \nabla f(X(t)), \dot{X}(t) \right\rangle \left( r(t) - \frac{\eta(t)W(t)}{w(t)} \right), \tag{3}$$

where we used the expressions for $\dot{Z}$ and $\nabla \psi^*(Z)$ from $\text{AMD}'_{w,\eta}$ in the second equality, and convexity of $f$ in the last inequality. Equipped with this bound, it becomes straightforward to give sufficient conditions for $L_r$ to be a Lyapunov function.

**Theorem 2.** *Suppose that for all $t \in [t_0, +\infty)$,*

1. *$\eta(t) \geq r'(t)$ and*
2. *$\left\langle \nabla f(X(t)), \dot{X}(t) \right\rangle \left( r(t) - \frac{\eta(t)W(t)}{w(t)} \right) \leq 0$.*

*Then $L_r$ is a Lyapunov function for $\text{AMD}_{w,\eta}$, and for all $t \geq t_0$, $f(X(t)) - f^\star \leq \frac{L_r(X(t_0), Z(t_0), t_0)}{r(t)}$.*

*Proof.* The two conditions, combined with inequality (3), imply that $\frac{d}{dt}L_r(X(t), Z(t), t) \leq 0$, thus $L_r$ is a Lyapunov function. Finally, since $D_{\psi^*}$ is non-negative, and $L_r$ is decreasing, we have

$$f(X(t)) - f^\star \leq \frac{L_r(X(t), Z(t), t)}{r(t)} \leq \frac{L_r(X(t_0), Z(t_0), t_0)}{r(t)}.$$

which proves the claim. $\square$

Note that the second condition depends on the solution trajectory $X(t)$, and may be hard to check a priori. However, we give one special case in which the condition trivially holds.

**Corollary 1.** *Suppose that for all $t \in [t_0, +\infty)$, $\eta(t) = \frac{w(t)r(t)}{W(t)}$, and $\frac{w(t)}{W(t)} \geq \frac{r'(t)}{r(t)}$. Then $L_r$ is a Lyapunov function for $\text{AMD}_{w,\eta}$, and for all $t \geq t_0$, $f(X(t)) - f^\star \leq \frac{L_r(X(t_0), Z(t_0), t_0)}{r(t)}$.*

Next, we describe a method to construct weight functions $w, \eta$ that satisfy the conditions of Corollary 1, given a desired rate $r$. Of course, it suffices to construct $w$ that satisfies $\frac{w(t)}{W(t)} \geq \frac{r'(t)}{r(t)}$, then to set $\eta(t) = \frac{w(t)r(t)}{W(t)}$. We can reparameterize the weight function by writing $\frac{w(t)}{W(t)} = a(t)$. Then integrating from $t_0$ to $t$, we have $\frac{W(t)}{W(t_0)} = e^{\int_{t_0}^{t} a(\tau)d\tau}$, and

$$w(t) = w(t_0)\frac{a(t)}{a(t_0)}e^{\int_{t_0}^{t} a(\tau)d\tau}. \tag{4}$$

Therefore the conditions of the corollary are satisfied whenever $w(t)$ is of the form (4) and $a : \mathbb{R}_+ \to \mathbb{R}_+$ is a continuous, positive function with $a(t) \geq \frac{r'(t)}{r(t)}$. Note that the expression of $w$ is defined up to the constant $w(t_0)$, which reflects the fact that the condition of the corollary is scale-invariant (if the condition holds for a function $w$, then it holds for $\alpha w$ for all $\alpha > 0$).

**Example 1.** *Let $r(t) = t^2$. Then $r'(t)/r(t) = 2/t$, and we can take $a(t) = \frac{\beta}{t}$ with $\beta \geq 2$. Then $w(t) = \frac{a(t)}{a(t_0)}e^{\int_{t_0}^{t} a(\tau)d\tau} = \frac{\beta/t}{\beta/t_0}e^{\beta \ln(t/t_0)} = (t/t_0)^{\beta-1}$ and $\eta(t) = \frac{w(t)r(t)}{W(t)} = \beta t$, and we recover the weighting scheme used in [11].*

**Example 2.** *More generally, if $r(t) = t^p$, $p \geq 1$, then $r'(t)/r(t) = p/t$, and we can take $a(t) = \frac{\beta}{t}$ with $\beta \geq p$. Then $w(t) = (t/t_0)^{\beta-1}$, and $\eta(t) = \frac{w(t)r(t)}{W(t)} = \beta t^{p-1}$.*

We also exhibit in the following a second energy function that is guaranteed to decrease under the same conditions. This energy function, unlike the Lyapunov function $L_r$, does not guarantee a specific convergence rate. However, it captures a natural measure of energy in the system. To define this energy function, we will use the following characterization of the inverse mirror map: By duality of the subdifferentials (e.g. Theorem 23.5 in [18]), we have for a pair of convex conjugate functions $\psi$ and $\psi^*$ that $x \in \partial \psi^*(x^*)$ if and only if $x^* \in \partial \psi(x)$. To simplify the discussion, we will assume that $\psi$ is also differentiable, so that $(\nabla \psi^*)^{-1} = \nabla \psi$ (this assumption can be relaxed). In what follows, we will denote by $\check{X} = \nabla \psi(X)$ and $\check{Z} = \nabla \psi^*(Z)$.

**Theorem 3.** *Let $(X(t), Z(t))$ be the unique maximal solution of $\mathrm{AMD}_{w,\eta}$, and let $\check{X} = \nabla\psi(X)$. Consider the energy function*

$$E_r(t) = f(X(t)) + \frac{1}{r(t)} D_{\psi^*}(Z(t), \check{X}(t)). \tag{5}$$

*Then if $w, \eta$ satisfy condition (2) of Theorem 2, $E_r$ is a decreasing function of time.*

*Proof.* To make the notation more concise, we omit the explicit dependence on time in this proof. We have $D_{\psi^*}(Z, \check{X}) = \psi^*(Z) - \psi^*(\check{X}) - \langle X, Z - \check{X}\rangle$. Taking the time-derivative , we have

$$\frac{d}{dt}D_{\psi^*}(Z, \check{X}) = \left\langle\nabla\psi^*(Z), \dot{Z}\right\rangle - \left\langle\nabla\psi^*(\check{X}), \dot{\check{X}}\right\rangle - \left\langle\dot{X}, Z - \check{X}\right\rangle - \left\langle X, \dot{Z} - \dot{\check{X}}\right\rangle$$

$$= \left\langle\nabla\psi^*(Z) - X, \dot{Z}\right\rangle - \left\langle\dot{X}, Z - \check{X}\right\rangle.$$

Using the second equation in $\mathrm{AMD}'_{w,\eta}$, we have $\nabla\psi^*(Z) - X = \frac{1}{a}\dot{X}$, and $\left\langle\dot{X}, Z - \check{X}\right\rangle = a\left\langle\nabla\psi^*(Z) - \nabla\psi^*(\check{X}), Z - \check{X}\right\rangle \geq 0$ by monotonicity of $\nabla\psi^*$. Combining, we have $\frac{d}{dt}D_{\psi^*}(Z, \check{X}) \leq -\frac{\eta}{a}\left\langle\dot{X}, \nabla f(X)\right\rangle$, and we can finally bound the derivative of $E_r$:

$$\frac{d}{dt}E_r(t) = \left\langle\nabla f(X), \dot{X}\right\rangle + \frac{1}{r}\frac{d}{dt}D_{\psi^*}(Z, \check{X}) - \frac{r'}{r^2}D_{\psi^*}(Z, \check{X})$$

$$\leq \left\langle\nabla f(X), \dot{X}\right\rangle\left(1 - \frac{\eta}{ar}\right).$$

Therefore condition (2) of Theorem 2 implies that $\frac{d}{dt}E_r(t) \leq 0$. $\qquad\square$

This energy function can be interpreted, loosely speaking, as the sum of a potential energy given by $f(X)$, and a kinetic energy given by $\frac{1}{r(t)}D_{\psi^*}(Z, \check{X})$: Indeed, when the problem is unconstrained, then one can take $\psi^*(z) = \frac{1}{2}\|z\|^2$, in which case $\nabla\psi^* = \nabla\psi = I$, the identity, and $D_{\psi^*}(Z, \check{X}) = \frac{1}{2}\|\check{Z} - X\|^2 = \frac{1}{2}\|\frac{\dot{X}}{a}\|^2$, a quantity proportional to the kinetic energy.

## 3 Primal Representation and Example Dynamics

An equivalent primal representation can be obtained by rewriting the equations in terms of $\check{Z} = \nabla\psi^*(Z)$ and its derivatives ($\check{Z}$ is a primal variable that remains in $\mathcal{X}$, since $\nabla\psi^*$ maps into $\mathcal{X}$). In this section, we assume that $\psi^*$ is twice differentiable on $E^*$. Taking the time derivative of $\check{Z}(t) = \nabla\psi^*(Z(t))$, we have

$$\dot{\check{Z}}(t) = \nabla^2\psi^*(Z(t))\dot{Z}(t) = -\eta(t)\nabla^2\psi^* \circ \nabla\psi(\check{Z}(t))\nabla f(X(t)),$$

where $\nabla^2\psi^*(z)$ is the Hessian of $\psi^*$ at $z$, defined as $\nabla^2\psi^*(z)_{ij} = \frac{\partial^2\psi^*(z)}{\partial z_j\partial z_i}$. Then using the averaging expression for $X$, we can write $\mathrm{AMD}_{w,\eta}$ in the following primal form

$$\mathrm{AMD}^p_{w,\eta}\begin{cases} \dot{\check{Z}}(t) = -\eta(t)\nabla^2\psi^* \circ \nabla\psi(\check{Z}(t))\nabla f\left(\frac{x_0 W(t_0) + \int_{t_0}^t w(\tau)\check{Z}(\tau)d\tau}{W(t)}\right) \\ \check{Z}(t_0) = x_0. \end{cases} \tag{6}$$

A similar derivation can be made for the mirror descent ODE without acceleration, which can be written as follows [11] (see also the original derivation of Nemirovski and Yudin in Chapter 3 in [13])

$$\mathrm{MD}\begin{cases} \dot{Z}(t) = -\nabla f(X(t)) \\ X(t) = \nabla\psi^*(Z(t)) \\ X(t_0) = x_0. \end{cases}$$

Note that this can be interpreted as a limit case of $\mathrm{AMD}_{\eta,w}$ with $\eta(t) \equiv 1$ and $w(t)$ a Dirac function at $t$. Taking the time derivative of $X(t) = \nabla\psi^*(Z(t))$, we have $\dot{X}(t) = \nabla^2\psi^*(Z(t))\dot{Z}(t)$, which leads to the primal form of the mirror descent ODE

$$\mathrm{MD}^p\begin{cases} \dot{X}(t) = -\nabla^2\psi^* \circ \nabla\psi(X(t))\nabla f(X(t)) \\ X(t_0) = x_0. \end{cases} \tag{7}$$

The operator $\nabla^2\psi^* \circ \nabla\psi$ appears in both primal representations (6) and (7), and multiplies the gradient of $f$. It can be thought of as a transformation of the gradient which ensures that the primal trajectory remains in the feasible set, this is illustrated in the supplementary material. For some choices of $\psi$, $\nabla^2\psi^* \circ \nabla\psi$ has a simple expression. We give two examples below.

We also observe that in its primal form, $\mathrm{AMD}^p_{w,\eta}$ is a generalization of the ODE family studied in [23], which can be written as $\frac{d}{dt}\nabla\psi(X(t) + e^{-\alpha(t)}\dot{X}(t)) = -e^{\alpha(t)+\beta(t)}\nabla f(X(t))$, for which they prove the convergence rate $\mathcal{O}(e^{-\beta(t)})$. This corresponds to setting, in our notation, $a(t) = e^{\alpha(t)}$, $r(t) = e^{\beta(t)}$ and taking $\eta(t) = a(t)r(t)$ (which corresponds to the condition of Corollary 1).

**Positive-orthant-constrained dynamics**   Suppose that $\mathcal{X}$ is the positive orthant $\mathbb{R}^n_+$, and consider the negative entropy function $\psi(x) = \sum_i x_i \ln x_i$. Then its dual is $\psi^*(z) = \sum_i e^{z_i - 1}$, and we have $\nabla\psi(x)_i = 1 + \ln x_i$ and $\nabla^2\psi^*(z)_{i,j} = \delta_i^j e^{z_i - 1}$, where $\delta_i^j$ is 1 if $i = j$ and 0 otherwise. Thus for all $x \in \mathbb{R}^n_+$, $\nabla^2\psi^* \circ \nabla\psi(x) = \mathrm{diag}(x)$. Therefore, the primal forms (7) and (6), reduce to, respectively,

$$\begin{cases}\forall i, \dot{X}_i = -X_i\nabla f(X)_i \\ X(0) = x_0\end{cases} \qquad\qquad \begin{cases}\forall i, \dot{\check{Z}}_i = -\eta(t)\check{Z}_i\nabla f(X)_i \\ \check{Z}(t_0) = x_0\end{cases}$$

where for the second ODE we write $X$ compactly to denote the weighted average given by the second equation of $\mathrm{AMD}_{w,\eta}$. When $f$ is affine, the mirror descent ODE lead to Lotka-Volterra equation which has applications in economics and ecology. For the mirror descent ODE, one can verify that the solution remains in the positive orthant since $\dot{X}$ tends to 0 as $X_i$ approaches the boundary of the feasible set. Similarly for the accelerated version, $\dot{\check{Z}}$ tends to 0 as $\check{Z}$ approaches the boundary, thus $\check{Z}$ remains feasible, and so does $X$ by convexity.

**Simplex-constrained dynamics: the replicator equation.**   Now suppose that $\mathcal{X}$ is the $n$-simplex, $\mathcal{X} = \Delta = \{x \in \mathbb{R}^n_+ : \sum_{i=1}^n x_i = 1\}$. Consider the distance-generating function $\psi(x) = \sum_{i=1}^n x_i \ln x_i + \delta_{\mathcal{X}}(x)$, where $\delta_{\mathcal{X}}(\cdot)$ is the convex indicator function of the feasible set. Then its conjugate is $\psi^*(z) = \ln\left(\sum_{i=1}^n e^{z_i}\right)$, defined on $E^*$, and we have $\nabla\psi(x)_i = 1 + \ln x_i$, $\nabla\psi^*(z)_i = \frac{e^{z_i}}{\sum_k e^{z_k}}$, and $\nabla^2\psi^*(z)_{ij} = \frac{\delta_i^j e^{z_i}}{\sum_k e^{z_k}} - \frac{e^{z_i}e^{z_j}}{\left(\sum_k e^{z_k}\right)^2}$. Then it is simple to calculate $\nabla^2\psi^* \circ \nabla\psi(x)_{ij} = \frac{\delta_i^j x_i}{\sum_k x_k} - \frac{x_i x_j}{\left(\sum_k x_k\right)^2} = \delta_i^j x_i - x_i x_j$. Therefore, the primal forms (7) and (6) reduce to, respectively,

$$\begin{cases}\forall i, \dot{X}_i + X_i\left(\nabla f(X)_i - \langle X, \nabla f(X)\rangle\right) = 0 \\ X(0) = x_0\end{cases} \quad \begin{cases}\forall i, \dot{\check{Z}}_i + \eta(t)\check{Z}_i\left(\nabla f(X)_i - \langle \check{Z}, \nabla f(X)\rangle\right) = 0 \\ \check{Z}(0) = x_0.\end{cases}$$

The first ODE is known as the replicator dynamics [19], and has many applications in evolutionary game theory [22] and viability theory [4], among others. See the supplementary material for additional discussion on the interpretation and applications of the replicator dynamics. This example shows that the replicator dynamics can be accelerated simply by performing the original replicator update on the variable $\check{Z}$, in which (i) the gradient of the objective function is scaled by $\eta(t)$ at time $t$, and (ii) the gradient is evaluated at $X(t)$, the weighted average of the $\check{Z}$ trajectory.

## 4   Adaptive Averaging Heuristic

In this section, we propose an adaptive averaging heuristic for adaptively computing the weights $w$. Note that in Corollary 1, we simply set $a(t) = \frac{\eta(t)}{r(t)}$ so that $\left\langle \nabla f(X(t)), \dot{X}(t)\right\rangle \left(r(t) - \frac{\eta(t)}{a(t)}\right)$ is identically zero (thus trivially satisfying condition (2) of Theorem 2). However, from the bound (3), if this term is negative, then this helps further decrease the Lyapunov function $L_r$ (as well as the energy function $E_r$). A simple strategy is then to adaptively choose $a(t)$ as follows

$$\begin{cases}a(t) = \frac{\eta(t)}{r(t)} & \text{if } \left\langle\nabla f(X(t)), \dot{X}(t)\right\rangle > 0, \\ a(t) \geq \frac{\eta(t)}{r(t)} & \text{otherwise.}\end{cases} \tag{8}$$

If we further have $\eta(t) \geq r'(t)$, then the conditions of Theorem 2 and Theorem 3 are satisfied, which guarantee that $L_r$ is a Lyapunov function and that the energy $E_r$ decreases. In particular, such a heuristic would preserve the convergence rate $r(t)$ by Theorem 2.

We now propose a discrete version of the heuristic when $r(t) = t^2$. We consider the quadratic rate in particular since in this case the discretization proposed by [11] preserves the quadratic rate, and corresponds to a first-order accelerated method[2] for which many heuristics have been developed, such as the restarting heuristics [17, 20] discussed in the introduction. To satisfy condition (1) of Theorem 2, we choose $\eta(t) = \beta t$ with $\beta \geq 2$. Note that in this case, $\frac{\eta(t)}{r(t)} = \frac{\beta}{t}$. In the supplementary material, we propose a discretization of the heuristic (8), using the correspondance $t = k\sqrt{s}$, for a step size $s$. The resulting algorithm is summarized in Algorithm 1, where $\psi^*$ is a smooth distance generating function, and $R$ is a regularizer assumed to be strongly convex and smooth. We give a bound on the convergence rate of Algorithm 1 in the supplementary material. The proof relies on a discrete counterpart of the Lyapunov function $L_r$.

The algorithm keeps $a_k = a_{k-1}$ whenever $f(\tilde{x}^{(k+1)}) \leq f(\tilde{x}^{(k)})$, and sets $a_k$ to $\frac{\beta}{k\sqrt{s}}$ otherwise. This results in a non-increasing sequence $a_k$. It is worth observing that in continuous time, from the expression (4), a constant $a(t)$ over an interval $[t_1, t_2]$ corresponds to an exponential increase in the weight $w(t)$ over that interval, while $a(t) = \frac{\beta}{t}$ corresponds to a polynomial increase $w(t) = (t/t_0)^{\beta-1}$. Intuitively, adaptive averaging increases the weights $w(t)$ on portions of the trajectory which make progress.

---

**Algorithm 1** Accelerated mirror descent with adaptive averaging

---

1: Initialize $\tilde{x}^{(0)} = x_0$, $\check{z}^{(0)} = x_0$, $a_1 = \frac{\beta}{\sqrt{s}}$
2: **for** $k \in \mathbb{N}$ **do**
3:     $\check{z}^{(k+1)} = \arg\min_{\check{z} \in \mathcal{X}} \beta k s \left\langle \nabla f(x^{(k)}), \check{z} \right\rangle + D_\psi(\check{z}, \check{z}^{(k)}).$
4:     $\tilde{x}^{(k+1)} = \arg\min_{\tilde{x} \in \mathcal{X}} \gamma s \left\langle \nabla f(x^{(k)}), \tilde{x} \right\rangle + R(\tilde{x}, x^{(k)})$
5:     $x^{(k+1)} = \lambda_{k+1}\check{z}^{(k+1)} + (1 - \lambda_{k+1})\tilde{x}^{(k+1)}$, with $\lambda_k = \frac{\sqrt{s}a_k}{1+\sqrt{s}a_k}$.
6:     $a_k = \min\left(a_{k-1}, \frac{\beta^{\max}}{k\sqrt{s}}\right)$
7:     **if** $f(\tilde{x}^{(k+1)}) - f(\tilde{x}^{(k)}) > 0$ **then**
8:         $a_k = \frac{\beta}{k\sqrt{s}}$

---

## 5   Numerical Experiments

In this section, we compare our adaptive averaging heuristic (in its discrete version given in Algorithm 1) to existing restarting heuristics. We consider simplex-constrained problems and take the distance generating function $\psi$ to be the entropy function, so that the resulting algorithm is a discretization of the accelerated replicator ODE studied in Section 3. We perform the experiments in $\mathbb{R}^3$ so that we can visualize the solution trajectories (the supplementary material contains additional experiments in higher dimension). We consider different objective functions: A strongly convex quadratic given by $f(x) = (x - s)^T A(x - s)$ for a positive definite matrix $A$, a weakly convex quadratic, a linear function $f(x) = c^T x$, and the Kullback-Leibler divergence, $f(x) = D_{\text{KL}}(x^\star, x)$. We compare the following methods:

1. The original accelerated mirror descent method (in which the weights follow a predetermined schedule given by $a_k = \frac{\beta}{k\sqrt{s}}$),
2. Our adaptive averaging, in which $a_k$ is computed adaptively following Algorithm 1,
3. The gradient restarting heuristic in [17], in which the algorithm is restarted from the current point whenever $\left\langle \nabla f(x^{(k)}), x^{(k+1)} - x^{(k)} \right\rangle > 0$,
4. The speed restarting heuristic in [20], in which the algorithm is restarted from the current point whenever $\|x^{(k+1)} - x^{(k)}\| \leq \|x^{(k)} - x^{(k-1)}\|$.

The results are shown in Figure 2. Each subfigure is divided into four plots: Clockwise from the top left, we show the value of the objective function, the trajectory on the simplex, the value of the energy function $E_r$ and the value of the Lyapunov function $L_r$.

The experiments show that adaptive averaging compares favorably to the restarting heuristics on all these examples, with a significant improvement in the strongly convex case. Additionally, the experiments confirm that under the adaptive averaging heuristic, the Lyapunov function is decreasing. This is not the case for the restarting heuristics as can be seen on the weakly convex example. It is interesting to observe, however, that the energy function $E_r$ is non-increasing for all the methods in our experiments. If we interpret the energy as the sum of a potential and a kinetic term, then this could be explained intuitively by the fact that restarting keeps the potential energy constant, and decreases the kinetic energy (since the velocity is reset to zero). It is also worth observing that even though the Lyapunov function $L_r$ is non-decreasing, it will not necessarily converge to 0 when there is more than one minimizer (its limit will depend on the choice of $z^\star$ in the definition of $L_r$).

Finally, we observe that the methods have a different qualitative behavior: The original accelerated method typically exhibits oscillations around the set of minimizers. The heuristics alleviate these oscillations in different ways: Intuitively, adaptive averaging acts by increasing the weights on portions of the trajectory which make the most progress, while the restarting heuristics reset the velocity to zero whenever the algorithm detects that the trajectory is moving in a bad direction. The speed restarting heuristic seems to be more conservative in that it restarts more frequently.

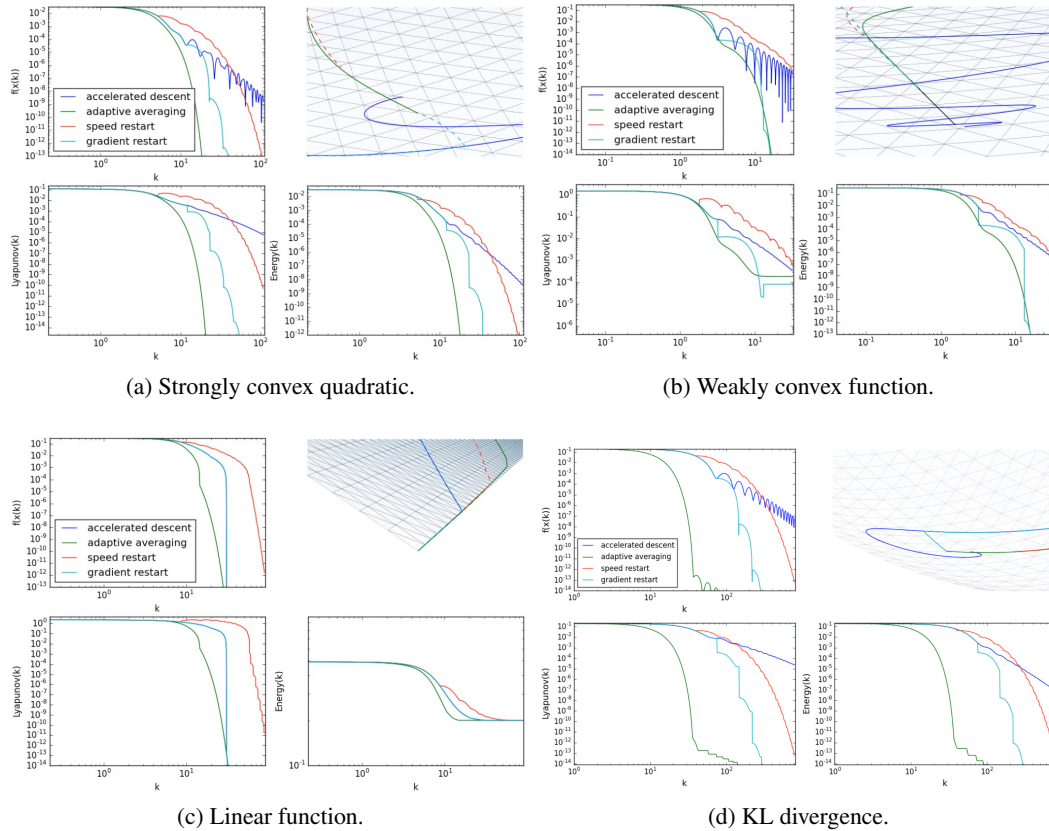

(a) Strongly convex quadratic.

(b) Weakly convex function.

(c) Linear function.

(d) KL divergence.

Figure 2: Examples of accelerated descent with adaptive averaging and restarting.

## 6 Conclusion

Motivated by the averaging formulation of accelerated mirror descent, we studied a family of ODEs with a generalized averaging scheme, and gave simple sufficient conditions on the weight functions to guarantee a given convergence rate in continuous time. We showed as an example how the replicator ODE can be accelerated by averaging. Our adaptive averaging heuristic preserves the convergence rate (since it preserves the Lyapunov function), and it seems to perform at least as well as other heuristics for first-order accelerated methods, and in some cases considerably better. This encourages further investigation into the performance of this adaptive averaging, both theoretically (by attempting to prove faster rates, e.g. for strongly convex functions), and numerically, by testing it on other methods, such as the higher-order accelerated methods proposed in [23].

## Footnotes

*Walid Krichene is currently affiliated with Google. walidk@google.com

[2]For faster rates $r(t) = t^p$, $p > 2$, it is possible to discretize the ODE and preserve the convergence rate, as proposed by Wibisono et al. [23], however this discretization results in a higher-order method such as Nesterov's cubic accelerated Newton method [16].

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
