[Supplementary Material]

# Adaptive Averaging in Accelerated Descent Dynamics
# Supplementary material, NIPS 2016

Walid Krichene        Alexandre Bayen        Peter Bartlett

## 1   Distance-Generating Functions and Mirror Operators

We consider a closed, convex set $\mathcal{X}$, and a pair of conjugate convex functions $\psi, \psi^*$ such that $\psi$ is closed and proper, and the effective domain of $\psi$ is $\mathcal{X}$. We denote $\mathcal{X}^*$ the effective domain of $\psi^*$. By Fenchel's duality theorem, $\psi^{**}$ coincides with $\psi$, and we have for all $x \in E$ and $z \in E^*$:

$$\psi^*(z) = \sup_{x \in E} \langle z, x \rangle - \psi(x), \qquad\qquad \psi(x) = \sup_{z \in E^*} \langle z, x \rangle - \psi^*(z).$$

Since $\psi$ and $\psi^*$ are proper convex functions, they are both subdifferentiable on the relative interior of their respective domains (Theorem 23.4 in [4]). And if we denote $\partial \psi(x)$ the subdifferential of $\psi$ at $x$, then we have, by definition of a subgrdient,

$$
\begin{aligned}
z \in \partial \psi(x) &\Leftrightarrow \psi(x') - \langle z, x' \rangle \geq \psi(x) - \langle z, x \rangle \ \forall x' \in E \\
&\Leftrightarrow x \in \arg\max_{x' \in E} \langle z, x' \rangle - \psi(x') \\
&\Leftrightarrow \psi^*(z) = \langle z, x \rangle - \psi(x)
\end{aligned}
$$

and switching the roles of $\psi$ and $\psi^*$ (and using the fact that $\psi^{**} = \psi$), we have the equivalence

$$\psi^*(z) + \psi(x) = \langle z, x \rangle \Leftrightarrow z \in \partial \psi(x) \Leftrightarrow x \in \partial \psi^*(z). \tag{1}$$

By the previous observation, we have for all $z \in E^*$,

$$\partial \psi^*(z) = \arg\max_{x \in E} \langle z, x \rangle - \psi(x). \tag{2}$$

And since $\operatorname{dom} \psi = \mathcal{X}$, we have that $\partial \psi^*(z) \subset \mathcal{X}$. Thus we have a set-valued function $\partial \psi^*(\cdot)$ which maps $E^*$ into $\mathcal{X}$. If $\psi^*$ is differentiable on all of $E^*$, then $\partial \psi^*(\cdot)$ becomes a (single-valued) function from $E^*$ to $\mathcal{X}$, as desired. The following proposition gives a sufficient condition for differentiability.

**Proposition 1.** *Let $\psi, \psi^*$ be a pair of convex, closed function which are conjugates of each other, and suppose that $\psi$ is strongly convex. Then $\psi^*$ is finite and differentiable on all of $E^*$.*

*Proof.* Strong convexity of $\psi$ implies, by Theorem 13.3 in [4], that $\operatorname{dom} \psi^* = E^*$, and by Theorem 25.3 in [4], that $\psi^*$ is essentially smooth (i.e. that it is differentiable on the interior of its domain, and that $\|\nabla \psi(x)\| \to \infty$ as $x$ approaches the boundary). But since $\operatorname{dom} \psi^* = E^*$, the interior of the domain is all of $E^*$. $\qquad\square$

## 2 Replicator Dynamics

The replicator ODE is given by

$$\begin{cases} \forall i, \ \dot{X}_i + X_i \left( \nabla_i f(X) - \langle X, \nabla f(X) \rangle \right) = 0 \\ X(0) = x_0 \end{cases}$$

It has been studied for a long time, see [5] for a survey, and has many applications ranging from evolutionary game theory [6] and viability theory [1] to transportation [2].

It is used to study large population dynamics, where one considers a population of players and a finite action set $\{1, \ldots, n\}$, such that at time $t$, $X_i(t)$ is the proportion of players who adopt action $i$. Then $\nabla_i f(X)$ is the cost (or the negative fitness) of action $i$ given the distribution $X$. The ODE is called replicator as it can be obtained using a simple model of adaptive play as follows: at time $t$, players are randomly matched in pairs, and if their current actions are, respectively, $i$ and $j$, then the first player will switch to $j$ (i.e. replicate the action of the second player) with a rate proportional to $\nabla_j f(X) - \nabla_i f(X)$, and similarly for the second player. As a consequence, the rate of increase of $X_i$ is simply the sum over all actions $j$ of $X_i X_j$ (the probability of the match $(i,j)$) multiplied by the difference in costs $\nabla_j f(X) - \nabla_i f(X)$, i.e.

$$\dot{X}_i = \sum_{j=1}^n X_i X_j (\nabla_j f(X) - \nabla_i f(X))$$

$$= X_i \left( \sum_{j=1}^n X_j (\nabla_j f(X) - \nabla_i f(X)) \right)$$

$$= X_i \left( \langle X, \nabla f(X) \rangle - \nabla_i f(X) \right).$$

## 3 Illustration of the operator $\nabla^2 \psi^* \circ \nabla \psi(Z)$

Consider the accelerated replicator dynamics given in the second example of Section 3. Recall that

$$\nabla^2 \psi^*(z)_{ij} = \frac{\delta_i^j e^{z_i}}{\sum_k e^{z_k}} - \frac{e^{z_i} e^{z_j}}{\left( \sum_k e^{z_k} \right)^2}.$$

And the primal version of $\mathrm{AMD}_{w,\eta}$ becomes

$$\begin{cases} \forall i, \dot{\check{Z}}_i + \eta(t) \check{Z}_i \left( \nabla_i f(X) - \langle \check{Z}, \nabla f(X) \rangle \right) = 0 \\ \check{Z}(0) = x_0. \end{cases}$$

This example can be used to illustrate the role of the Hessian term in equation (6). Suppose that $\nabla \psi^*(Z)$ approaches the relative boundary of the feasible set, say $e^{Z_{i_0}}$ approaches 0. Then $(\nabla^2 \psi^*(Z) \nabla f(X))_{i_0} = \frac{e^{Z_{i_0}}}{\sum_k e^{Z_k}} \left( \nabla_{i_0} f(X) - \left\langle \nabla f(X), \frac{e^Z}{\sum_e Z_k} \right\rangle \right)$, also approaches 0. Figure 1 displays the vector field $\nabla^2 \psi^*(Z) \nabla f(X)$ for different values of $Z$, to illustrate this phenomenon.

(a) Vector field $\nabla^2\psi^*(Z(t_1))\nabla f(\cdot)$.

(b) Vector field $\nabla^2\psi^*(Z(t_2))\nabla f(\cdot)$.

Figure 1: Vector field $X \mapsto \nabla^2\psi^*(Z(t))\nabla f(X)$ for different values of $Z(t)$ (taken along a solution trajectory for an example problem with solution on the relative boundary of the simplex). As $\nabla\psi^*(Z(t))$ approaches the relative boundary, the vector field changes in such a way that the component that is orthogonal to the boundary vanishes.

# 4   Discretization of $\mathrm{AMD}'_{w,\eta}$

Starting from the ODE with generalized averaging,

$$\mathrm{AMD}'_{w,\eta}\begin{cases} \dot{Z}(t) = -\eta(t)\nabla f(X(t)) \\[4pt] \dot{X}(t) = \frac{w(t)}{W(t)}(\nabla\psi^*(Z(t)) - X(t)) \\[4pt] X(t_0) = \nabla\psi^*(Z(t_0)) = x_0, \end{cases}$$

we apply a discretization similar to that used in [3]. Let the step size be $\sqrt{s}$, and apply a mixed Euler scheme (forward in the $Z$ variable, and backward in the $X$ variable). Given a solution $(X, Z)$ of the ODE, let $t_k = k\sqrt{s}$, and $x^{(k)} = X(t_k) = X(k\sqrt{s})$. Approximating $\dot{X}(t_k)$ with $\frac{X(t_k + \sqrt{s}) - X(t_k)}{\sqrt{s}}$, and $\dot{Z}(t_k)$ with $\frac{Z(t_k + \sqrt{s}) - Z(t_k)}{\sqrt{s}}$, we can write

$$\begin{cases} \frac{z^{(k+1)} - z^{(k)}}{\sqrt{s}} = -\eta_k \nabla f(x^{(k)}), \\[4pt] \frac{x^{(k+1)} - x^{(k)}}{\sqrt{s}} = a_{k+1}\left(\nabla\psi^*(z^{(k+1)}) - x^{(k+1)}\right). \end{cases} \tag{3}$$

with $\eta_k := \eta(k\sqrt{s})$ and $a_k := a(k\sqrt{s})$. The second equation can be rewritten as

$$x^{(k+1)} = \left(x^{(k)} + \sqrt{s}a_{k+1}\nabla\psi^*(z^{(k+1)})\right) / \left(1 + \sqrt{s}a_{k+1}\right).$$

Thus, $x^{(k+1)}$ is a convex combination of $\nabla\psi^*(z^{(k)})$ and $x^{(k)}$ with coefficients $\lambda_{k+1} = \frac{\sqrt{s}a_{k+1}}{1+\sqrt{s}a_{k+1}}$ and $1-\lambda_{k+1} = \frac{1}{1+\sqrt{s}a_{k+1}}$.

Next, using the characterization of the mirror operator $\nabla\psi^*$, given in equation (2), we can write that

$$
\begin{aligned}
\check{z}^{(k+1)} &= \nabla\psi^*(z^{(k+1)}) \\
&= \operatorname*{arg\,min}_{x\in\mathcal{X}} \psi(x) - \left\langle z^{(k+1)}, x \right\rangle \\
&= \operatorname*{arg\,min}_{x\in\mathcal{X}} \psi(x) - \left\langle \nabla\psi(\check{z}^{(k)}) - \sqrt{s}\eta_k\nabla f(x^{(k)}), x \right\rangle \\
&= \operatorname*{arg\,min}_{x\in\mathcal{X}} \sqrt{s}\eta_k \left\langle \nabla f(x^{(k)}), x \right\rangle + D_\psi(x, \check{z}^{(k)})
\end{aligned}
$$

To summarize, the discretization can be written purely in terms of the primal variables $x^{(k)}$ and $\check{z}^{(k)}$ as follows

$$
\begin{cases}
x^{(k+1)} = \lambda_{k+1}\check{z}^{(k+1)} + (1-\lambda_{k+1})x^{(k)}, \ \lambda_k = \frac{\sqrt{s}a_k}{1+\sqrt{s}a_k}, \\
\check{z}^{(k+1)} = \operatorname{arg\,min}_{x\in\mathcal{X}} \sqrt{s}\eta_k \left\langle \nabla f(x^{(k)}), x \right\rangle + D_\psi(x, \check{z}^{(k)}).
\end{cases} \tag{4}
$$

Now to preserve the quadratic convergence rate, we show in [3] show that it suffices to replace, in the expression (4) of $x^{(k+1)} = \lambda_{k+1}\nabla\psi^*(z^{(k+1)}) + (1-\lambda_{k+1})x^{(k)}$, the term $x^{(k)}$ with $\tilde{x}^{(k)}$, obtained as a solution to the following regularized problem

$$
\tilde{x}^{(k+1)} = \operatorname*{arg\,min}_{x\in\mathcal{X}} \gamma s \left\langle \nabla f(x^{(k)}), x \right\rangle + R(x, x^{(k)}),
$$

where $R$ is regularization function that is both strongly convex and smooth. After applying this modification, we have the following algorithm:

---
**Accelerated mirror descent**
---
1: Initialize $\tilde{x}^{(0)} = x_0$, $\check{z}^{(0)} = x_0$,
2: **for** $k \in \mathbb{N}$ **do**
3: $\quad \check{z}^{(k+1)} = \operatorname{arg\,min}_{\check{z}\in\mathcal{X}} \sqrt{s}\eta_k \left\langle \nabla f(x^{(k)}), \check{z} \right\rangle + D_\psi(\check{z}, \check{z}^{(k)})$.
4: $\quad \tilde{x}^{(k+1)} = \operatorname{arg\,min}_{\tilde{x}\in\mathcal{X}} \gamma_k s \left\langle \nabla f(x^{(k)}), \tilde{x} \right\rangle + R(\tilde{x}, x^{(k)})$
5: $\quad x^{(k+1)} = \lambda_{k+1}\check{z}^{(k+1)} + (1-\lambda_{k+1})\tilde{x}^{(k+1)}$, with $\lambda_k = \frac{\sqrt{s}a_k}{1+\sqrt{s}a_k}$.
6: **end for**
---

Finally, $a_k$ can be computed adaptively as described in Section 4, which results in Algorithm 1, copied below.

---
**Algorithm 1** Accelerated mirror descent with adaptive averaging. Parameters $\beta^{\max} \geq \beta \geq 3$ and step size $s$.
---
1: Initialize $\tilde{x}^{(0)} = x_0$, $\check{z}^{(0)} = x_0$, $a_1 = \frac{\beta}{\sqrt{s}}$
2: **for** $k \in \mathbb{N}$ **do**
3: $\quad \check{z}^{(k+1)} = \operatorname{arg\,min}_{\check{z}\in\mathcal{X}} \beta k s \left\langle \nabla f(x^{(k)}), \check{z} \right\rangle + D_\psi(\check{z}, \check{z}^{(k)})$.
4: $\quad \tilde{x}^{(k+1)} = \operatorname{arg\,min}_{\tilde{x}\in\mathcal{X}} \gamma s \left\langle \nabla f(x^{(k)}), \tilde{x} \right\rangle + R(\tilde{x}, x^{(k)})$
5: $\quad x^{(k+1)} = \lambda_{k+1}\check{z}^{(k+1)} + (1-\lambda_{k+1})\tilde{x}^{(k+1)}$, with $\lambda_k = \frac{\sqrt{s}a_k}{1+\sqrt{s}a_k}$.
6: $\quad a_k = \min(a_{k-1}, \frac{\beta^{\max}}{k\sqrt{s}})$.
7: $\quad$ **if** $f(\tilde{x}^{(k+1)}) > f(\tilde{x}^{(k)})$ **then**
8: $\quad\quad a_k = \frac{\beta}{k\sqrt{s}}$.
9: $\quad$ **end if**
10: **end for**
---

Note that the conditions on $a_k$ ensures that $\frac{\beta}{k\sqrt{s}} \leq a_k \leq \frac{\beta^{\max}}{k\sqrt{s}}$, which will be important in proving the convergence rate of the discretization.

We will show that the sequence

$$
\tilde{L}^{(k)} := L_r(\tilde{x}^{(k)}, z^{(k)}, k\sqrt{s}) = sk^2(f(\tilde{x}^{(k)}) - f^\star) + D_{\psi^*}(z^{(k)}, z^\star)
$$

is a Lyapunov function for the discrete dynamics given in Algorithm 1. In the following, we will suppose that $\beta^{\max} \geq \beta \geq 3$, that $\psi^*$ is $L_{\psi^*}$ smooth, and that $R$ is $\ell_R$ strongly convex and $L_R$ smooth.

**Lemma 1.** *If $\gamma \geq \beta\beta^{\max}L_R L_{\psi^*}$ and $s \leq \frac{\ell_R}{2L_f\gamma}$, then*

$$\tilde{L}^{(k+1)} - \tilde{L}^{(k)} \leq (2k + 1 - k\beta)sf(\tilde{x}^{(k+1)}) - f^\star.$$

*It follows that $\tilde{L}^{(k)}$ is non-increasing for $k \geq 1$.*

The proof is given below, and is an extension of the proof of Lemma 2 in [3]. As a consequence, we can prove that adaptive averaging preserves the quadratic convergence rate.

**Theorem 1.** *Accelerated mirror descent with adaptive averaging, given in Algorithm 1, with step sizes $\gamma \geq \beta\beta^{\max}L_R L_{\psi^*}$ and $s \leq \frac{\ell_R}{2L_f\gamma}$, guarantees that for all $k > 0$,*

$$f(\tilde{x}^{(k)}) - f^\star \leq \frac{\tilde{L}^{(1)}}{sk^2}.$$

*Proof.* Since $\tilde{L}^{(k)}$ is non-increasing for all $k \geq 1$, we have

$$f(\tilde{x}^{(k)}) - f^\star \leq \frac{\tilde{L}^{(k)}}{sk^2} \leq \frac{\tilde{L}^{(1)}}{sk^2}.$$

$\square$

In the proof of Lemma 1, we will use the following lemmas, which can be found in [3].

**Lemma 2** (Convexity inequality). *Let $f$ be a convex function and suppose that $\nabla f$ is $L_f$-Lipschitz w.r.t. $\|\cdot\|$. Then for all $x, x', x^+$,*

$$f(x^+) \leq f(x') + \langle \nabla f(x), x^+ - x' \rangle + \frac{L_f}{2}\|x^+ - x\|^2$$

**Lemma 3** (Bregman identity). *For all $u$, $v$, $w$*

$$D_{\psi^*}(u, v) - D_{\psi^*}(w, v) = -D_{\psi^*}(w, u) + \langle \nabla\psi^*(u) - \nabla\psi^*(v), u - w \rangle$$

**Lemma 4** (Bregman bound). *For all $u, v \in E^*$,*

$$\frac{1}{2L_{\psi^*}}\|\check{u} - \check{v}\|^2 \leq D_{\psi^*}(u, v) \leq \frac{L_{\psi^*}}{2}\|u - v\|_*^2$$

*where $\check{u} = \nabla\psi^*(u)$ and $\check{v} = \nabla\psi^*(v)$.*

*Proof of Lemma 1.* We start by bounding the difference in Bregman divergences

$$
\begin{aligned}
&D_{\psi^*}(z^{(k+1)}, z^\star) - D_{\psi^*}(z^{(k)}, z^\star) \\
&\quad = -D_{\psi^*}(z^{(k)}, z^{(k+1)}) + \left\langle \nabla\psi^*(z^{(k+1)}) - \nabla\psi^*(z^\star), z^{(k+1)} - z^{(k)} \right\rangle \quad \text{by Lemma 3,} \\
&\quad \leq -\frac{1}{2L_{\psi^*}}\|\check{z}^{(k+1)} - \check{z}^{(k)}\|^2 + \left\langle \check{z}^{(k+1)} - x^\star, -\beta k s \nabla f(x^{(k)}) \right\rangle \quad\quad \text{by Lemma 4.}
\end{aligned}
$$

$$\tag{5}$$

Now using the step from $x^{(k)}$ to $\tilde{x}^{(k+1)}$, we have

$$\tilde{x}^{(k+1)} = \underset{x \in \mathcal{X}}{\arg\min}\, \gamma s \left\langle \nabla f(x^{(k)}), x \right\rangle + R(x, x^{(k)})$$

with $\frac{\ell_R}{2}\|x-y\|^2 \leq R(x, y) \leq \frac{L_R}{2}\|x-y\|^2$. Therefore, for any $x$, $R(x, x^{(k)}) \geq R(\tilde{x}^{(k+1)}, x^{(k)}) + \gamma s \left\langle \nabla f(x^{(k)}), \tilde{x}^{(k+1)} - x \right\rangle$. We can write

$$\check{z}^{(k+1)} - \check{z}^{(k)} = \frac{1}{\lambda_k}\left(\lambda_k \check{z}^{(k+1)} + (1 - \lambda_k)\tilde{x}^{(k)} - x^{(k)}\right) = \frac{1}{\lambda_k}\left(d^{(k+1)} - x^{(k)}\right),$$

where we have defined $d^{(k+1)}$ in the obvious way. Thus

$$\|\check{z}^{(k+1)} - \check{z}^{(k)}\|^2$$

$$= \frac{1}{\lambda_k^2}\|d^{(k+1)} - x^{(k)}\|^2$$

$$\geq \frac{1}{\lambda_k^2}\frac{2}{L_R}R(d^{(k+1)}, x^{(k)})$$

$$\geq \frac{1}{\lambda_k^2}\frac{2}{L_R}\left(R(\tilde{x}^{(k+1)}, x^{(k)}) + \gamma s\left\langle \nabla f(x^{(k)}), \tilde{x}^{(k+1)} - d^{(k+1)}\right\rangle\right)$$

$$\geq \frac{1}{\lambda_k^2}\frac{2}{L_R}\left(\frac{\ell_R}{2}\|\tilde{x}^{(k+1)} - x^{(k)}\|^2 + \gamma s\left\langle \nabla f(x^{(k)}), \tilde{x}^{(k+1)} - \lambda_k\check{z}^{(k+1)} - (1-\lambda_k)\tilde{x}^{(k)}\right\rangle\right).$$

Thus, multiplying by $\frac{\lambda_k\beta kL_R}{2\gamma}$,

$$\frac{\lambda_k\beta kL_R}{2\gamma}\|\check{z}^{(k+1)} - \check{z}^{(k)}\|^2$$

$$\geq \frac{\beta k\ell_R}{2\lambda_k\gamma}\|\tilde{x}^{(k+1)} - x^{(k)}\|^2 + \left\langle \beta ks\nabla f(x^{(k)}), \frac{1}{\lambda_k}\tilde{x}^{(k+1)} - \check{z}^{(k+1)} - \frac{1-\lambda_k}{\lambda_k}\tilde{x}^{(k)}\right\rangle. \quad (6)$$

Subtracting (6) from (5),

$$D_{\psi^*}(z^{(k+1)}, z^\star) - D_{\psi^*}(z^{(k)}, z^\star)$$

$$\leq -\alpha_k\|\check{z}^{(k+1)} - \check{z}^{(k)}\|^2 - \frac{\beta k\ell_R}{2\lambda_k\gamma}\|\tilde{x}^{(k+1)} - x^{(k)}\|^2$$

$$+ \left\langle -\beta ks\nabla f(x^{(k)}), -x^\star + \frac{1}{\lambda_k}\tilde{x}^{(k+1)} - \frac{1-\lambda_k}{\lambda_k}\tilde{x}^{(k)}\right\rangle,$$

where

$$\alpha_k = \frac{1}{2L_{\psi^*}} - \frac{\beta k\lambda_k L_R}{2\gamma}.$$

Defining $D_1^{(k+1)} = \|\tilde{x}^{(k+1)} - x^{(k)}\|^2$ and $D_2^{(k+1)} = \|\check{z}^{(k+1)} - \check{z}^{(k)}\|^2$, we can rewrite the last inequality as

$$D_{\psi^*}(z^{(k+1)}, z^\star) - D_{\psi^*}(z^{(k)}, z^\star)$$

$$= -\alpha_k D_2^{(k+1)} - \frac{\beta k\ell_R}{2\lambda_k\gamma}D_1^{(k+1)} + \beta sk\left\langle -\nabla f(x^{(k)}), \tilde{x}^{(k+1)} - x^\star\right\rangle$$

$$+ \frac{1-\lambda_k}{\lambda_k}\beta sk\left\langle -\nabla f(x^{(k)}), \tilde{x}^{(k+1)} - \tilde{x}^{(k)}\right\rangle$$

By Lemma 2, we can bound the inner products as follows

$$\left\langle \tilde{x}^{(k+1)} - \tilde{x}^{(k)}, -\nabla f(x^{(k)})\right\rangle \leq f(\tilde{x}^{(k)}) - f(\tilde{x}^{(k+1)}) + \frac{L_f}{2}D_1^{(k+1)},$$

$$\left\langle \tilde{x}^{(k+1)} - x^\star, -\nabla f(x^{(k)})\right\rangle \leq f^* - f(\tilde{x}^{(k+1)}) + \frac{L_f}{2}D_1^{(k+1)}.$$

Combining the last inequalities,

$$D_{\psi^*}(z^{(k+1)}, z^\star) - D_{\psi^*}(z^{(k)}, z^\star)$$

$$\leq -\alpha_k D_2^{(k+1)} - \frac{\beta k\ell_R}{2\lambda_k\gamma}D_1^{(k+1)} + \beta ks\left(f^\star - f(\tilde{x}^{(k+1)}) + \frac{L_f}{2}D_1^{(k+1)}\right)$$

$$+ \beta ks\frac{1-\lambda_k}{\lambda_k}\left(f(\tilde{x}^{(k)}) - f(\tilde{x}^{(k+1)}) + \frac{L_f}{2}D_1^{(k+1)}\right)$$

$$= \beta ks\frac{1-\lambda_k}{\lambda_k}\left(f(\tilde{x}^{(k)}) - f(\tilde{x}^{(k+1)})\right) + \beta ks\left(f^\star - f(\tilde{x}^{(k+1)})\right) - \alpha_k D_2^{(k+1)} - \beta_k D_1^{(k+1)},$$

where

$$\beta_k = \frac{\beta k\ell_R}{2\lambda_k\gamma} - \frac{\beta ksL_f}{2} - \frac{\beta ksL_f}{2}\frac{1-\lambda_k}{\lambda_k} = \frac{\beta k}{2\lambda_k}\left(\frac{\ell_R}{\gamma} - L_f s\right).$$

Next, observe that $\frac{1-\lambda_k}{\lambda_k} = \frac{1}{\sqrt{s}a_k}$, and by construction of $a_k$ (lines 6–8 in Algorithm 1), we have $\frac{1}{a_k\sqrt{s}}(f(\tilde{x}^{(k+1)}) - f(\tilde{x}^{(k)})) \leq \frac{k}{\beta}(f(\tilde{x}^{(k+1)}) - f(\tilde{x}^{(k)}))$, so

$$\frac{1-\lambda_k}{\lambda_k}(f(\tilde{x}^{(k+1)}) - f(\tilde{x}^{(k)})) = \frac{1}{\sqrt{s}a_k}(f(\tilde{x}^{(k+1)}) - f(\tilde{x}^{(k)})) \leq \frac{k}{\beta}(f(\tilde{x}^{(k+1)}) - f(\tilde{x}^{(k)})).$$

Combining with the previous inequality, we have

$$D_{\psi^*}(z^{(k+1)}, z^\star) - D_{\psi^*}(z^{(k)}, z^\star)$$
$$\leq k^2 s\left(f(\tilde{x}^{(k)}) - f(\tilde{x}^{(k+1)})\right) + \beta k s\left(f^* - f(\tilde{x}^{(k+1)})\right) - \alpha_k D_2^{(k+1)} - \beta_k D_1^{(k+1)},$$

Finally, we obtain a bound on the difference $\tilde{L}^{(k+1)} - \tilde{L}^{(k)}$:

$$\tilde{L}^{(k+1)} - \tilde{L}^{(k)}$$
$$= (k+1)^2 s(f(\tilde{x}^{(k+1)}) - f^\star) - k^2 s(f(\tilde{x}^{(k)}) - f^\star) + D_{\psi^*}(z^{(k+1)}, z^\star) - D_{\psi^*}(z^{(k)}, z^\star)$$
$$= k^2 s(f(\tilde{x}^{(k+1)}) - f(\tilde{x}^{(k)})) + (2k+1)s(f(\tilde{x}^{(k+1)}) - f^\star) + D_{\psi^*}(z^{(k+1)}, z^\star) - D_{\psi^*}(z^{(k)}, z^\star)$$
$$\leq (2k+1 - \beta k)s(f(\tilde{x}^{(k+1)}) - f^\star) - \alpha_k D_2^{(k+1)} - \beta_k D_1^{(k+1)}$$

For the desired inequality to hold, it suffices that $\alpha_k, \beta_k \geq 0$, i.e.

$$\frac{1}{2L_{\psi^*}} - \frac{\beta k \lambda_k L_R}{2\gamma} \geq 0$$
$$\frac{\beta k}{2\lambda_k}\left(\frac{\ell_R}{\gamma} - L_f s\right) \geq 0,$$

i.e.

$$\gamma \geq \beta k \lambda_k L_R L_{\psi^*}$$
$$s \leq \frac{\ell_R}{L_f \gamma}.$$

To simplify the condition on $\gamma$, we observe that $\lambda_k = \frac{1}{1 + \frac{1}{\sqrt{s}a_k}}$, and since $a_k \leq \frac{\beta^{\max}}{k\sqrt{s}}$, we have

$$\beta k \lambda_k \leq \frac{\beta k}{1 + \frac{k}{\beta^{\max}}} \leq \beta \beta^{\max}$$

So it is sufficient that

$$\gamma \geq \beta \beta^{\max} L_R L_{\psi^*} \qquad\qquad s \leq \frac{\ell_R}{L_f \gamma}$$

which concludes the proof. □

## 5    Additional Numerical Experiments

We provide additional numerical experiments in higher dimension $n = 100$, to illustrate the performance of the adaptive averaging compared to the restarting heuristics. We test the algorithm on simplex-constrained problems, with quadratic objective functions $f(x) = (x - s)^T A(x - s)$ with a positive definite matrix $A$ in the first example, and a positive semidefinite matrix in the second example (with rank 10), a linear function in the third example, and the Kullback Leibler divergence in the last example. The results are reported in Figure 2. Each subfigure has three plots: From left to right, we show the value of objective function, the Lyapunov function and the energy function. We observe similar results to those in dimension 3. Adaptive averaging performs as well as the restarting heuristics, it gives a significant improvement in one of the examples (in this case the linear example), and it guarantees the decrease of the Lyapunov function.

(a) Strongly convex quadratic.

(b) Weakly convex function.

(c) Linear function.

(d) KL divergence.

Figure 2: Examples of accelerated descent with adaptive averaging and restarting on simplex constrained problems.