[Reviews · NeurIPS 2016]

Reviewer 1

Summary

The paper leverages the ODE interpretation of Nesterov's acceleration of Krichene et al. in NIPS 2015 to design and evaluate a natural heuristic way to adaptively reweigh the iterates of the algorithm to speed up convergence. In contrast with previous adapative averaging heuristics, this method is guaranteed to dominate the standard fixed-schedule averaging. The empirical evaluation shows that the method yields speed-ups on a general class of functions.

Qualitative Assessment

I enjoyed reading this paper. The mathematical argument is simple and easy to follow, especially given knowledge of the work of Krichene et al. At the same time, the empirical results show very promising speed-ups. While the technical development is a close extension of Krichene et al, I believe that this paper has a large potential impact, because Nesterov's acceleration is so pervasive in application. Too many times, even within empirical evaluations, a new, possibly highly adaptive method is compared to Nesterov's acceleration with a fixed weight sequence. However, as this paper confirms, rendering Nesterov's method adaptive only requires a small modification and can lead to dramatic improvements. To improve the paper from a technical point of view, I would recommend providing an analysis of the discretized process showing that the rate of convergence is preserved. If I am not mistaken, this should be relatively straightforward. I would also be interested in possible connections between the adaptive averaging of Nesterov's method and the Conjugate Gradient method for strongly convex quadratic optimization.

Confidence in this Review

2-Confident (read it all; understood it all reasonably well)


Reviewer 2

Summary

Building upon an early work that connects accelerated Mirror Descent and continuous-time ODE, the authors extend the ODE dynamics with a more general averaging scheme and propose a novel adaptive averaging strategy for one of its discretization. In some cases, the new variant of AMD with adaptive averaging outperforms existing AMD algorithms with restarting.

Qualitative Assessment

The proposed adaptive averaging strategy is interestingly new and comes with a good intuition. While the paper looks nice overall, the originality and usefulness remains a big concern. - Most elements (e.g. Lyapunov function, discretization, etc.) directly follow from an early NIPS paper [7]. The extension from simple averaging to generalized averaging is kind of trivial and not of much use since eventually the authors only consider the case with quadratic rate; faster rate would require higher-order methods which are not always applicable. - Some results, e.g., interpretation of energy function, reformulation into primal form, reduction to replicator dynamics, promote a better understanding of accelerated Mirror Descent algorithm, but are not significantly important to advance machine learning. - Although the adaptive averaging is introduced to satisfy the conditions sufficient to guarantee convergence of ODE, it is not fully theoretically grounded due to the approximation during discretization. Still, there is no explicit convergent result provided, meaning it is again just a heuristic, as referred by the authors. Also, there exist a number of different averaging schemes of Nester's accelerated algorithms; without comparing to other alternatives, it is unclear whether this adaptive variant makes a big different. - The experiment on a toy example with only quadratic/linear objective is not interesting or convincing enough to demonstrate the adavantage with adaptive averaging, especially for the general case with strongly convexity. == post-rebuttal answer== I have read the authors' rebuttal and they promise to add - additional higher dimensional and non-quadratic examples in the revision - theoretical proof of convergence of adaptive averaging - implementation of adaptive averaging on Nesterov's cubic-regulariezed Newton method - discussion comparing to other adaptive averaging I have updated my scores accordingly.

Confidence in this Review

2-Confident (read it all; understood it all reasonably well)


Reviewer 3

Summary

The paper studies accelerated first order methods for constrained convex optimization in the continuous and discrete setting. The authors continues the study initiated in by Krichene et al. in [7], where the Nesterov's method is shown to be an appropriate discretization of an ODE system, coupling a primal trajectory with a dual one. In particular, the primal trajectory is obtained performing a specific averaging of the mirror of the dual variable. In the present paper, more general averaging schemes are studied. Theorem 2 then derives convergence rates, under suitable conditions on the averaging coefficients. In the continuous setting, convergence rates that are faster w,r,too 1/t^2 are derived. An adaptive choice of the weights with convergence guarantees is also studied and then used in the experimental section. Such choice is compared with restarting techniques, which are known to improve convergence of accelerated methods in practice. The theoretical analysis of the discrete version of the algorithm is not included.

Qualitative Assessment

I think this is an interesting and well-written paper. The analysis of the new averaging scheme for accelerated methods is presented in the continuous setting and is rigorous. Even if the main ideas of the analysis of the continuous dynamics are now well-established in a series of papers, there are new contributions and results. In this respect, I think that the work by Attouch, Peypouquet and coauthors should be mentioned. The main theorem, showing that there are averaging schemes with faster convergence rate than 1/t^2 is a relevant one, even if it would be interesting to understand if this improved rate is preserved when passing to the discrete procedure.

Confidence in this Review

3-Expert (read the paper in detail, know the area, quite certain of my opinion)


Reviewer 4

Summary

This paper studies a class of ODEs with a generalized averaging scheme. Sufficient conditions on the dual learning rate and the weights are given to achieve a desired convergence rate using a Lyapunov argument. Moreover, the author(s) use(s) an adaptive averaging heuristic to speed up the decrease of the Lyapunov function. Both theoretical guarantees and numerical comparison are provided.

Qualitative Assessment

The is a long line of works aiming to empirically speed up the convergence of accelerated methods. This paper makes a crucial observation via ODEs that adaptively averaging the weights along the solution trajectories can also achieve the speedup. I found this new method very interesting.

Confidence in this Review

2-Confident (read it all; understood it all reasonably well)


Reviewer 5

Summary

The paper studies accelerated descent dynamics for constrained convex optimization. By using Lyapunov function, the authors are able to achieve desired convergence rate. An adaptive averaging heuristic is designed to speed up the convergence rate of Lyapunov. Results show that the adaptive averaging performs equally well or even better than adaptive restarting.

Qualitative Assessment

I have little knowledge in this field. However, the idea that using the adaptive averaging to achieve a desired faster convergence rate for constrained convex optimization seems interesting and promising. Quality: no comments Novelty: no comments Impact: no comments Clarity: no comments

Confidence in this Review

1-Less confident (might not have understood significant parts)


Reviewer 6

Summary

This paper presents an adaptive averaging heuristic for accelerating constrained convex optimization. The presented method provides the guarantee that it preserves the original convergence rate when the objective function is not strongly convex. This is an advantage over existing restarting heuristics and the paper shows this both theoretically and experimentally.

Qualitative Assessment

I had a hard time to follow the content of the paper as it is out of my area of expertise. I assume that the used abbreviations and terms are known to those readers that are addressed by the paper, but I would suggest to include background information. The equations between lines 33,34 and 46,47 could benefit from a more intuitive explanation. Furthermore, I would like to know the relation to adaptive averaging methods used in non-convex optimization such as Adagrad [1] or Adam [2]. Finally, I am wondering about the practical significance of this work. The experiments were carried out in R^3 which is obviously a toyish setup. How does your method (and related work) scale to much higher dimensions? Minor comment: Reference to solid line in Figure 1 is ambiguous. Maybe use a dotted line or colors to show the trajectory of the primal variable? [1] Duchi, John, Elad Hazan, and Yoram Singer. "Adaptive subgradient methods for online learning and stochastic optimization." Journal of Machine Learning Research 12.Jul (2011): 2121-2159. [2] Kingma, Diederik, and Jimmy Ba. "Adam: A method for stochastic optimization." International Conference on Learning Representations (ICLR). (2014).

Confidence in this Review

1-Less confident (might not have understood significant parts)